# From Bacterial Poisons to Toxins: The Early Works of Pasteurians

**DOI:** 10.3390/toxins14110759

**Published:** 2022-11-03

**Authors:** Jean-Marc Cavaillon

**Affiliations:** National Research Agency (ANR), 75012 Paris, France; jean-marc.cavaillon@pasteur.fr

**Keywords:** infection, poison, serotherapy, soluble bacterial chemicals, vaccine

## Abstract

We review some of the precursor works of the Pasteurians in the field of bacterial toxins. The word “toxin” was coined in 1888 by Ludwig Brieger to qualify different types of poison released by bacteria. Pasteur had identified the bacteria as the cause of putrefaction but never used the word toxin. In 1888, Émile Roux and Alexandre Yersin were the first to demonstrate that the bacteria causing diphtheria was releasing a deadly toxin. In 1923, Gaston Ramon treated that toxin with formalin and heat, resulting in the concept of “anatoxin” as a mean of vaccination. A similar approach was performed to obtain the tetanus anatoxin by Pierre Descombey, Christian Zoeller and G. Ramon. On his side, Elie Metchnikoff also studied the tetanus toxin and investigated the cholera toxin. His colleague from Odessa, Nikolaï GamaleÏa who was expected to join Institut Pasteur, wrote the first book on bacterial poisons while other Pasteurians such as Etienne Burnet, Maurice Nicolle, Emile Césari, and Constant Jouan wrote books on toxins. Concerning the endotoxins, Alexandre Besredka obtained the first immune antiserum against lipopolysaccharide, and André Boivin characterized the biochemical nature of the endotoxins in a work initiated with Lydia Mesrobeanu in Bucharest.

## 1. From Bacteria of Putrefaction to Putrid Poisons, Bacterial Metabolites and Toxins

Peter Ludvig Panum (1820–1885), a Danish physician and physiopathologist working in Copenhagen was among the very first scientists in 1856 to consider that putrefaction was associated with a putrid poison. It took him a while before he recognized that such poison could derive from the bacteria of putrefaction [1,2]. Efforts were then made to characterize these poisons, and Ernst von Bergmann (1836–1907), a Baltic German surgeon identified what he called sepsin [3]. The word toxin was coined in 1888 by Ludwig Brieger (1849–1919), a professor of medicine at Humboldt University in Berlin [4] who had identified putrescine and cadaverin, organic compounds released by bacteria of putrefaction (1885). In 1878, Francesco Selmi (1817–1881), an Italian chemist from Bologna, coined the word ptomaïnes, from Greek ptōma (corpse), to name the cadaveric alkaloids [5]. Many years later, Elie Metchnikoff (1845–1916) who worked at the Institut Pasteur as soon it had been inaugurated [6,7], also reported that bacteria of putrefaction were producing poisons [8]. He identified bacterial metabolites released by gut bacteria [9]. Two of them (indol and paracresol) were rendered responsible of tissue lesions and vasculature alterations when injected in rabbits and could even induce death when injected into monkeys.

Two other scientists who trained at Institut Pasteur made a significant contribution in the field. Sir Marc Armand Ruffer (1859–1917) who later on became the father of paleopathology [10] worked with Pasteur and Metchnikoff from 1889 to 1890. While in Paris, he also worked with Albert Charrin (1856–1907) at the school of medicine and made a key observation [11]: the filtered supernatants of bacteria (*Pseudomonas aeruginosa*), devoid of any alive or dead bacteria could induce fever once injected into rabbits. Eugenio Centanni, an italian pathologist, named the responsible bacterial poison ‘pyrotoxina’ [12].

Accordingly, they had identified the effects of endotoxin, three years before Richard Pfeiffer (1858–1945) developed his concept of endotoxin [13]. Most fascinatingly, the authors suspected that fever was the consequence of the activation of macrophages, far before the discoveries of endogenous pyrogens (interleukin-1, IL-1; IL-6, etc.) were made [14]. The second one, Nikolaï Gamaleïa (1859–1949), attended the inaugural ceremony of Institut Pasteur, and was expected to join the institute [7]. In 1886, he was sent by the city of Odessa where he had worked with his mentor, Elie Metchnikoff, to spend four months with Pasteur’s team to learn the preparation procedures of the anthrax and rabies vaccines. Pasteur sent him to London to follow the works of the British investigative committee on rabies vaccine. From 1886 to 1892, Gamaleïa shared his life between Odessa and Paris. In 1892, he published his investigations on the cholera poisons [15], in which he described two deleterious substances, one he called «nucléine» present in heated (120 °C) culture media of *Vibrio cholerae*, highly toxic for guinea-pigs, rabbits, pigeons and dogs. The second substance, a nucleo-albumin, obtained after filtration through Chamberland filter, was shown to be heat sensitive and to induce all cholera symptoms, including diarrhea. The same year Gamaleïa further characterized the diphtheria toxin [16]. He showed that both pepsin and trypsin destroyed the diphtheria poison. Still in 1892, Gamaleïa published the first treaty on bacterial poisons [17] (translated in English the following year). He considered that a new science has sprung up: the science of microbial poisons, which is based at once on bacteriology, on biological chemistry, and on general physiology. In his book Gamaleïa addresses all the emerging aspects of this new specialty. After offering a historical background, he argued that infectious diseases were an intoxication by the poisons of the pathogenic microbes, considering that the poisons are intimately linked to the bodies of the bacteria. After addressing the chemical nature of the bacterial poisons, he summarized the knowledges on tetanus, diphtheria, cholera, tuberculosis and anthrax, while referring to the early works on immunity against these toxins, particularly the antisera generated against diphtheria and tetanus toxins.

## 2. The Soluble Chemical Products Produced by Bacteria and Immunity

Louis Pasteur was aware of the works of Panum and von Bergman, although he mentioned to have failed to reproduce the work of the later [18]. Pasteur did not seem to have been initially fascinated by the concept of bacterial poison. In 1880, he wrote: “*Speak, if you want, of poisoning. Make this hypothesis, I accept it. I do not know the mechanism of death from any disease more than you or anyone else does, no more than we know the mechanism of life. Speak of poison, if you wish, but you will be forced to add that, if a poison causes death, it is the microbe that generates the poison*” [19]. Indeed, he rarely mentioned the word poison and never used the word toxin. However, while working on fowl cholera, in 1880, he acknowledged that a soluble substance was released by the vibrio: “*I have acquired that during the life of the parasite, a narcotic is made, and that it is this narcotic which provokes the morbid symptom so pronounced of sleep in fowl cholera*” [20]. Later, Pasteur realized that these soluble substances could both induce the disease and be used to generate immunity: “*The fine memoir of MM. Roux and Chamberland, contained in the December 1887 issue of the Annales of M. Duclaux, demonstrates with perfect rigor that the life of the septic vibrio develops soluble chemical products which act on it little by little like an antiseptic. Introduced in sufficient quantity into the body of guinea pigs, these products give them immunity to the fatal disease caused by this vibrio. The proof is thus made, that immunity, against a disease so serious and so quickly mortal, can be obtained by the injection of chemical substances which can be measured, and that these substances themselves result from the life of the deadly microbes. […] The first, among the observers who have occupied themselves with this subject, I had sought to produce immunity in hens by means of the soluble products formed in a broth of culture by the life of the microbe of the fowl cholera. I saw the symptoms of the disease to appear, but not the immunity; which was perhaps, as MM. Roux and Chamberland observed, only a question of the quantity of soluble products used in my experience*” [21]. Despite Pasteur recognized that immunity could be achieved with bacterial soluble substances, as stated by Kendal Smith: “*a careful reading of Pasteur’s presentations to the Academy of Sciences reveals that Pasteur was entirely mistaken as to how immunity occurs, in that he reasoned, as a good microbiologist would, that appropriately attenuated microbes would deplete the host of vital trace nutrients absolutely required for their viability and growth, not an active response on the part of the host*” [22]. Indeed, Roux and Chamberland opposed themselves to their master: “*M. Pasteur thought that in the case of fowl cholera, the non-recurrence was due to the disappearance of some substances consumed by the microbe.*” When they recognized that “*It is not necessary for the cells of the pathogenic organism to live among the cells of the animal to confer immunity on it*” [23]. On his side, Auguste Chauveau (1827–1917), a French veterinarian, director of the National veterinary school of Alfort disagreed with Pasteur’s definition of immunity: “*Also M. Pasteur believed himself more and more authorized to consider the organism as a medium of culture which, by a first attack of the disease, would lose, under the influence of the culture of the parasite, principles that life would not return there or would return there only after a certain time*” [24]. However, the idea of Chauveau was not anymore correct: “*immunity is due to a substance left in the body by the culture of the microbe and which opposes its further development*.” To be opposed to the Master was not an easy task: “*This judgment, passed by a master such as M. Pasteur, was of a nature to discourage opposing convictions, even the most robust. I hesitated for a moment to keep mine. Nevertheless, as, meanwhile, new experiments had come to fully confirm my first observations, it was impossible for me not to support the deductions which I had drawn from them. I therefore maintained them with respectful firmness*” [24].

## 3. Diphtheria Toxin

Known as “the strangling angel of children,” diphtheria was a deadly bacterial infection killing thousands of babies and children every year. The germ, *Corynebacterium diphtheria*, was discovered in Germany in 1883 by Edwin Klebs (1834–1913), isolated and cultured the following year by Friedrich Loeffler (1852–1915). From 1888 to 1890, Alexandre Yersin (1863–1943) and Émile Roux (1853–1933) (Figure 1) who was the third director of Institut Pasteur after Louis Pasteur and Émile Duclaux (1840–1904), undertook their investigations on diphtheria [25]. They studied filtrates of 42 days cultures of the bacterium obtained through the porcelain filter set up by Charles Chamberland (1851–1908), one of the closest collaborators of Pasteur. Once injected in guinea pigs, the filtrates ended to the death of the animals, while rabbits and pigeons were more resistant than guinea pigs. Their analyses allowed them to claim that a toxin was elaborated by the germ of diphtheria. Furthermore, they also found the toxin in urines collected from sick children shortly before their deaths. In August 1891, E. Roux had the opportunity to precise his view during the seventh International *Congress of Hygiene and Demography* in London [26]. “*It is natural to conclude that microbes act through their chemical products, true poisons specific to each of them, and which determine the symptoms of the disease in man and in animals […] The infectious disease is therefore a poisoning: the source of the poison is the microbe settled in the tissues; it elaborates his toxin there at the expense of the living being that it is going to kill*”. *Furthermore, he addressed a new way* to induce protective immunity, he called “chemical vaccination” in contrast to the microbial vaccination defined by Pasteur in the case of fowl cholera and by Henry Toussaint (1847–1890) in the case of anthrax: “*It is therefore not necessary for the refractory state to be acquired, that the microbes penetrate into the body, it suffices that the substances prepared in the artificial cultures be introduced into it. If, therefore, by swarming in the organism, microbes give immunity, it is undoubtedly because they produce the same chemicals that we find in* in vitro *cultures*” [26].

In 1890, the use of immune sera against diphtheria toxin and tetanus toxin was rendered popular by Emil von Behring (1854–1917) and Shibasaburo Kitasato (1853–1931) who offered the basis of serotherapy to treat diphtheria and tetanus [27]. Von Behring and Kitasato prepared immune sera in guinea-pigs, rabbits, sheeps, goats, and horses. Adolf Baginsky (1843–1918) reported the first clinical trial in 220 children with diphtheria who received intraperitoneal injection of immune serum, ending to a healing rate of 77% [28]. Paul Ehrlich (1854–1915), the father of humoral immunity who shared the Nobel prize in 1908 with Metchnikoff, used the anti-toxin responsiveness to demonstrate the transmission of immunity during pregnancy and suckling [29] and developed a standardized method allowing to ensure reproducible titers of high toxin neutralizing activity in the antiserum [30].

Of note, two years before von Behring, two French physicians and scientists, friends since secondary school, Jules Héricourt (1850–1938) and Charles Richet (1850–1935) reported that the peritoneal transfusion of whole blood of dogs previously inoculated with *Staphylococcus pyosepticus* into rabbits was able to transfer immunity in these rabbits once challenged with the same bacteria, in contrast to the absence of protection provided by the blood from uninfected dogs [31]. Their failure to provide protection with a similar approach in the case of tuberculosis and cancer led to the oblivion of their pioneer works [32]. Later, Richet was awarded the Nobel prize 1913 for his discovery of anaphylaxis, and Emil von Behring was awarded with the very first Nobel prize in medicine or physiology in 1901. In his Nobel lecture, von Behring paid tribute to his precursors: “*Without the preliminary works by Loeffler and Roux, there would be no serum treatment for diphtheria.*”

In addition to have discovered the toxin, to prepare the immune sera, Émile Roux developed the use of horses on a large scale with the help of Edmond Nocard (1850–1903). Nocard had joined Pasteur’s laboratory in 1880 and was Director of the veterinary school of Alfort. Horses were first hosted at the veterinary school before a large number of horses could join the stables of the annex of Institut Pasteur in Marnes-La-Coquette (Figure 2). With an initial number of a dozen in September 1894, 136 horses were hosted by the beginning of 1895. Because the Parisian production remained insufficient to cover the need of the whole country, institutions were created in various cities (including Institut Pasteur de Lille, and production centers in Lyon, Le Havre, Grenoble, Bordeaux, Marseille, Rouen). In Nancy, the center was created thanks to a donation of Osiris (1825–1907), a generous donator who had regularly supported the Institut Pasteur [33].

In 1894, Émile Roux with Louis Martin (1864–1946), a former student of Joseph Grancher (1843–1907), his colleague of Institut Pasteur, reported with great details the preparation of the diphtheria toxin and the immune sera [34]. It contrasted with the vagueness maintained by Pasteur on the preparation of his vaccine against anthrax, which was at odds with the scientific attitude of the very rigorous Dr. Roux. With Dr. Auguste Chaillou (1866–1915), from Necker Hospital, they published the treatment of 300 children, ending with a 50% decreased mortality [35]. The official announcement of the success of the experiment was made by Roux on 5 September 1894 in Budapest during the eighth International Congress of Hygiene and Demography (Figure 3). The results had a great impact in the lay press (Figure 4). “Le Figaro” of which the director was Gaston Calmette (1858–1914), the brother of the Pasteurian Albert Calmette (1863–1933), echoed this success and launched for a call for donations that helped to collect more than 612,000 francs and to gather numerous retired horses. Sarah Bernardt, a star actress, also raised funds for the Institut Pasteur, offering a hundred seats up for auction, at the theater de la Renaissance, for the 1894 première of Gismonda.

However, because Roux had compared the outcome of patients treated with immune serum in one hospital with that of untreated patients in another hospital, some physicians were not fully convinced. Among those, were Johannes A. G. Fibiger (1867–1928), a junior physician, working in the ward of Søren Thorvald Sørensen (1849–1928) at Blegdamshospitalet in Copenhagen. In 1896–1897, Fibiger conducted the very first randomized controlled trial. Treatment allocation depended on the day of admittance, and patients received either standard treatment or standard treatment plus serotherapy. Only patients for whom the diphtheria bacterium was then identified were kept in the study. Eight out of 239 patients (3.3%) in the serum treated group, and 30 out of 245 (12.2%) in the control group died [36,37]. Of note, later, Fibiger was awarded with a Nobel Prize (1926), for his discovery of “*Spiroptera carcinoma*” (*Gongylonema neoplasticum*, its current name), a nematode to which he attributed a role in the development of gastric cancer in rats. Systematic reanalyses of Fibiger’s data led to conclude that his specific finding was found erroneous [38], despite the links between certain pathogens and cancer were then fully recognized.

Gaston Ramon (1886–1963) (Figure 5) was a veterinarian, hired by Roux to prepare the horse antisera at the annex of Institut Pasteur. In 1923, he showed that the treatment of the diphtheria toxin with formalin and heat resulted in an immunizing molecule that had lost its toxicity, but retained its antigenicity as shown in guinea pigs and horses [39]. Ramon coined the word anatoxin. The same year Alexander Thomas Glenny (1882–1965) from the Wellcome Research Laboratories, Beckenham (Kent) called toxoid his preparation that was only obtained after formalin treatment and needed to be mixed with anti-toxin to be used as an immunizing agent [40]. In France, the vaccination of infants against diphtheria with the Ramon anatoxin has been compulsory since the law of 24 June 1938. Among the other major contributions of Ramon, let us mention the discovery of the adjuvants: “*It is necessary to involve an inflammatory reaction at the site of antigen injection to enhance the immune response*.” [41], the famous «dirty little secret of the immunologists» as claimed by Charles Janeway (1943–2003), the father of the rebirth of innate immunity. Ramon had observed that if a small abscess was occurring at the site of injection of his anatoxin, the antibody titer was greatly enhanced. Then, on purpose he created an inflammation by mixing different substances with his vaccine, including tapioca, his favorite one. Gaston Ramon has been nominated 155 folds for the Nobel prize and is #1 of the scientists with the greatest number of nominations without being awarded. Of note, with 115 nominations Émile Roux is #2 on this podium!

## 4. Tetanus Toxin

The bacillus of tetanus (*Clostridium tetani*) was first described in 1884 in Berlin by Arthur Nicolaier (1862–1942). In 1890, three groups reported the presence of a released toxin: In Denmark, Knud Faber (1862–1956), chief physician at Frederiks Hospital and later at Rigshospitalet and professor of clinical medicine at the University of Copenhagen [42]; In Italy, at the University of Bologna, Giuseppina Cattani (1859–1914) and Guido Tizzoni (1853–1932), medical doctors and bacteriologists [43] and Alessandro Bruschettini (1868–1932) [44]; and in France, Louis Vaillard (1850–1935), and Hyacinthe Jean Vincent (1862–1950) both professors at the military medical school (Val-de-Grâce) showed that filtered cultures of the tetanus bacillus could induce the disease and kill mice, guinea pigs and rabbits [45]. Émile Roux and Louis Vaillard, following the precursor works of von Behring and Kitasato [27]; initiated a collaboration to further define the preparation of the toxin and its treatment with iodine to obtain an appropriate less toxic immunogen to generate in mice’s, rabbit’s, guinea-pig’s, sheep’s, cow’s and horse’s protective immune sera [46]. They showed that in rabbits the protective immunity lasted more than two years, although they advocated to perform regular boosts. Most interestingly, they reported that the cow milk was a source of anti-toxin in agreement with the ingenious experiments of Ehrlich [29]. In agreement with Metchnikoff’s observations, they suggested that the immune sera act on leukocytes to favor the phagocytosis of the tetanus germs, a phenomenon known as opsonization. They showed that the immune sera could be protective even when administered after the deadly bacteria. They also report their first attempts in humans with horse immune sera. Five patients died but two survived. Of course, an inappropriate timing may explain these results. The death of an eleven-year-old child after the extraction of two teeth reminds us that by the end of the 19th century, tetanus could happen in unexpected settings. In 1896, Dr. Eugene Tracey, a British physician reported how he saved a little girl with tetanus by injecting her the immune serum of Drs. Cattani & Tizzoni [47]. In 1897, Élie Metchnikoff compared the sensitivity of different animal species to the tetanus toxin and their respective capacity to produce anti-toxin. He reported that scorpions, beetles, carps, axolotl, tortoise were insensitive while hens, guinea-pigs, frog and caiman when maintained at 32–37 °C, were producing anti-toxin [48]. Auguste-Charles Marie (1864–1935), working in Elie Metchnikoff’s laboratory localized the toxin after its injection within the blood compartment, the nervous system and in other organs of frogs, guinea-pigs, rabbits, mice and dogs [49].

Edmond Nocard played again a major role to favor the use of horses for tetanus serotherapy [50]. Institut Pasteur was engaged in a central role during world war I to provide antisera to the armies. In 1914, the Institut Pasteur had 300 horses and was preparing 80,000 vials of antisera a month. In 1918 there were 1462 horses, allowing the preparation of 600,000 vials per month.

After the successful approach by Ramon to prepare the diphtheria anatoxin, the same experimental methodology was developed in 1925 to prepare the tetanus anatoxin, by Pierre Descombey (1895–1930), a young pasteurian who passed away when he was only 35. He successfully immunized horses and guinea pigs [51]. The following year after Albert Lafaille (1891–1963), Ramon’s colleague had tested the safety of the injection on himself, Ramon and his colleague Christian Joseph Zoeller (1888–1934), professor at the Val de Grâce military hospital, tested the tetanus anatoxin on a hundred humans and obtained strong neutralizing activity of their sera after three injections [52]. In France, a law of 15 August 1936 rendered the anti-tetanus vaccination mandatory within the armies, and the association of both anatoxins was officialized in 1940 for children after Ramon and Zoeller had shown the feasibility of such association [53].

## 5. Cholera Toxin

In 1886, The Neapolitan physician, Antonio Cantani, was the first to presume that the cholera toxin was linked to the body substance of the bacteria [54]—a concept which had been further developed by others, especially by Gamaleïa in 1892 [15].

At Institut Pasteur, Metchnikoff compared the sensitivity of different animal species to *Vibrio cholera*. He found that guinea pigs were more sensitive than rabbits, which were more sensitive than mice, while pigeons and hens were insensitive [55]. In addition, he demonstrated that the toxicity was due to a toxin and that the antitoxin immunity was protective. For his demonstration he placed live bacteria in a bag preventing the bacteria’s dispersal, and then he implanted the bag into the peritoneal cavity of guinea pigs. Most of them died, thus illustrating that a diffusible product was responsible of the poor outcome. When he placed dead bacteria or culture medium in the bag, the animals survived. Then, he selected the few animals that survived the bags containing the live bacteria and re-injected them with a lethal dose of *V. cholerae*. Not only the animals were protected, but they were able to resist further injections of up to 16 lethal doses. Afterwards, Metchnikoff prepared the cholera toxin and successfully immunized guinea pigs, rabbits, goats and horses demonstrating that their sera displayed a protective activity. It is admirable that the father of cellular immunity made key experiments demonstrating the importance of humoral immunity against bacterial toxins.

Despite these precursors works, the discovery of the cholera toxin is often attributed to an Indian investigator from Calcutta, Sambhu Nath De (1915–1985) who reported in 1959 the toxicity of bacteria-free culture filtrate of *Vibrio cholerae* in rabbits [56]. He showed that the cholera toxin could kill the rabbit, inducing a fall of blood pressure, heart edema, an increased permeability of the capillaries of the intestinal mucosa and an alteration of the kidneys.

## 6. Endotoxins

Richard Pfeiffer (1858–1945) coined the word endotoxin in 1892 [13]. Pfeiffer had worked as Robert Koch’s assistant in the Institute of Hygiene in Berlin. In 1894, he reported the process of in vivo bacteriolysis (1894), known as “the Pfeiffer phenomenon” which was further deciphered by Jules Bordet at Institut Pasteur [57,58]. Pfeiffer accompanied Koch in 1897 in India to investigate the plague epidemics. Pfeiffer thought that the bacteria were containing such a toxic substance, before it was recognized that endotoxins are present on the surface of Gram-negative bacteria and are regularly released by growing bacteria [59]. Two major achievements were obtained by Pasteurians in the field of endotoxins.

Alexandre Besredka (1870–1940) was born in Odessa and had Metchnikoff as a teacher. He moved to Paris in 1893 to study medicine and joined Metchnikoff’s laboratory. At his master’s death in 1916, Besredka, having spent more than 20 years at his side, became the head of the laboratory Metchnikoff had founded at Institut Pasteur. While Pfeiffer was mistaken in his appreciation of the inability of endotoxins to induce neutralizing antibodies, it was Alexandre Besredka who made the decisive discovery that antisera raised against intravenously injected bacteria developed endotoxin-neutralizing properties. A horse injected with typhoid vaccine generated antibodies against typhoid endotoxin that were able to protect guinea-pigs injected with 30 lethal doses of endotoxin. Once again, it is quite fascinating that in the laboratory of the father of innate cellular immunity, such a key experiment in the field of humoral immunity had been achieved [60].

André Boivin (1895–1949) joined the Institut Pasteur annex in Garches in 1936, working with Gaston Ramon, pursuing his research on smooth and rough endotoxins he had initiated together with Ion and Lydia Mesrobeanu, while he was at the Cantacuzene Institute in Bucharest [61,62]. Boivin and Mesrobeanu had deciphered the biochemical nature of the endotoxins, they initially called “antigène glucido-lipidique”, before it acquired his popular name of lipopolysaccharide (LPS).

## 7. The First Pasteurian Books on Toxins

In 1911, Etienne Burnet (1873–1960) published a book entitled «Microbes and Toxins» (translated in English in 1912) with a preface of Metchnikoff. Burnet was a physician who first joined the École Normale Supérieure (ENS) where Pasteur made most of his career. He entered at Institut Pasteur in 1904 as an assistant in the laboratory of Amédée Borrel (1867–1936) and from 1907–1919, he worked in the laboratory of Metchnikoff. His book is not only a book of bacteriology, dealing with the nature of microbes, but he also addressed the host response, immunity, inflammation, phagocytosis, anaphylaxis, and vaccines. In his chapter devoted to the toxins, he not only focused on bacterial toxins, but he also mentioned the vegetal ones, including ricin. Of note, Jan Danysz (1860–1928), a biologist from Poland who worked with Jules Bordet in South Africa, successfully proposing serotherapy to fight bovine plague [58], investigated the interaction of ricin with its anti-toxin antibodies [63]. Burnet compared the soluble toxins with diastases (enzymes) and also described the endotoxins, paying tribute to Besredka who was the first to report anti-endotoxins.

The second book to be mentioned has been published in 1919. Entitled “Toxines et antitoxines”, it has never been translated in English. It addresses vegetal toxins, those found in animal venoms and produced by bacteria, emphasizing on the pathophysiological consequences of their injections through different routes in experimental animals, and the acquired immunity associated with the protective role of antibodies. It was written by three authors: Maurice Nicolle (1862–1932), Emile Césari (1876–1956) and Constant Jouan (1877–1949). Maurice Nicolle is the oldest brother of Charles Nicolle (1866–1936) who received the Nobel prize in 1928 for his work performed while he was director of the Institut Pasteur in Tunis, demonstrating that lice transmit typhus. In 1893, Maurice Nicolle was appointed to replace Waldemar Haffkine (1860–1930) who had developed the cholera and the plague vaccines [64], as preparator of the microbiology course at the Institute. Soon, Louis Pasteur sent him to Turkey at the request of Sultan Abdul Hamid II. There, he directed the Imperial Institute of Bacteriology of Constantinople. From 1896 to 1915, Maurice Nicolle reported numerous investigations on the preparation of toxins, their preservation, and the properties of antisera. Back at the Institut Pasteur Institute (1902–1926), he attracted a number of young researchers and students around him, including Emile Césari who became his assistant in 1909. Their work on toxins and antitoxins, especially on the mutual precipitation of antigens and antibodies, allowed them to develop a method for measuring the in vitro activity of diphtheria and tetanus toxins on the one hand, and on the other hand, the antitoxic potency of anti-diphtheria and anti-tetanus sera. At the request of E. Roux and A. Calmette, he became head of the anti-venomous serotherapy department. The third co-author, Constant Jouan started in 1893 as a preparator in the Department of microbiology applied to hygiene and vaccinations, directed by Charles Chamberland. In 1909, he became assistant laboratory head, responsible for the manufacture of vaccines under the responsibility of Emile Roux. He shared this function with Ernest Fernbach (1875–1962) and came in charge of the laboratory of anthrax vaccines. In 1909, he registered a patent for an apparatus allowing the centrifugation of liquids. In 1919 when war had just ended in Europe, Constant Jouan, decided to devote all of his time to the creation of a new laboratory instruments company. His ingenuity and creativity ensured the rapid growth of the Jouan company. The first to adapt an electric motor to a centrifuge (until then manually operated), he opened new perspectives in research and diversified towards chemistry and biology, making a consortium with other older companies (Maison Adnet (Paris, France) for sterilization and hospital devices; and Maison Mathieu (Paris, France) for surgical instruments). The Nantes (France)-based Jouan company was acquired in September 2003 by its major competitor, the American Thermo Electron.

## 8. The French-German Relationships

The war of 1870–1871 between France and the Prussian Empire profoundly traumatized Louis Pasteur. He left Paris, worrying about a possible destruction of his laboratory at the École Normale Supérieure during the siege of Paris. He moved to Arbois, then to Pontarlier, Geneva, Lyon and finally joined Émile Duclaux in Clermond-Ferrand. There, he worked in the close by city of Chamalières, at the Kuhn breweries to improve the local production of beers. In June 1871, he patented his process to prepare and preserve beers. In his text, one finds the famous sentence: “*I wish that the beers manufactured with my process carry in France the name of Beer of the National Revenge*”. He sent back his “Honoris causa” degree offered by the University of Bonn and refused to be awarded with the Prussian order of the Royal Crown proposed by the emperor, Guillaume of Prussia (Kronenorden). His words “*science has no homeland, but the scientist has one*” were recalled by Paul Brouardel (1837–1906), the Dean of the Faculty of Medicine during his Jubilé. To his former student, Jules Raulin (1836–1896) Pasteur wrote: “*I try to keep away all these memories and the sight of all our miseries to which I see no salvation except in the despair of an all-out struggle. I would like France to resist until her last man, until her last rampart! I would like the war to be prolonged until the heart of winter so that, the elements coming to our aid, all these vandals would perish from cold, misery and disease. Each of my works until my last day will bear for epigraph: Hate to Prussia. Revenge. Revenge*.” To Luigi Chiozza (1828–1889) a chemist and Italian deputy, in response to an invitation to leave France and to join Italy, Pasteur wrote: “*My dear friend, I received your second letter. How touched I am by these steps and how happy I would be without the misfortunes of my country of these testimonies of esteem granted to my works, I who have always lived for glory.*” These last words are a most interesting confession of his motivation. Of course, his hate of Prussia, after the humiliating defeat and the loss of Alsace, a place where he had been professor in his early career (teaching chemistry in Strasbourg), was shared by many of his country people. Among those was Jean-Jacques Henner (1829–1905), an Alsatian artist and friend of his family. He was famous for his allegorical painting “L’Alsace, is waiting” (1871). In the following years, Henner painted the portrait of his daughter, Marie-Louise Pasteur (1876), his daughter-in-law, Jeanne Pasteur (1854–1932), wife of Jean-Baptiste Pasteur (1877), and of Louis Pasteur himself (1877).

Despite his Germanophobic attitude, Pasteur had little influence over his close collaborators. In 1888, Émile Roux sent Alexandre Yersin to visit the laboratory of Robert Koch, Pasteur’s best enemy, to follow the courses and to bring back ideas of organization for the laboratories and for teaching purposes. In 1895, Joseph Grancher in his preface of the re-edition of the book “Etude sur les virus” by Jean Hameau (1779–1851) recognized the superiority of the German school in terms of bacterial staining, bacterial cultures on semi-solid media and regarding their identifications [65]. In 1904, Metchnikoff welcomed Robert Koch while he visited the Institut Pasteur. Roux, Metchnikoff and von Behring were friends to the point that Roux and Metchnikoff were the godfathers of Behring’s sons. On 4 February 1914, few months before Germany declared war to France, the Pasteurians received Paul Ehrlich and his wife, offering a historical picture of both Nobel prize 1908 standing close to each other (Figure 6). Finally, in March 1914, Roux and Metchnikoff paid tribute to the work of Paul Ehrlich in a paper published in French in a German journal [66]. The complementarity of the works of the Pasteurians and of the German school was illustrated all along this period, when key discoveries in bacteriology and immunology were made on both side of the Rhein.

## Figures and Tables

**Figure 1 toxins-14-00759-f001:**
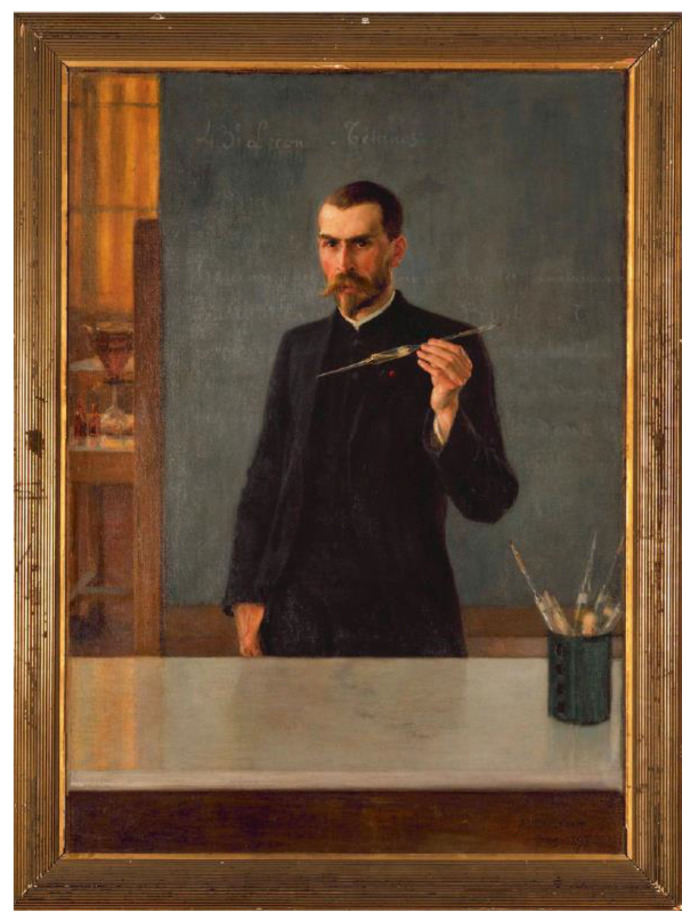
Portrait of Émile Roux giving his serotherapy course. Oil on canvas by the Finnish painter Albert Edelfelt (1895). ©Institut Pasteur/Musée Pasteur.

**Figure 2 toxins-14-00759-f002:**
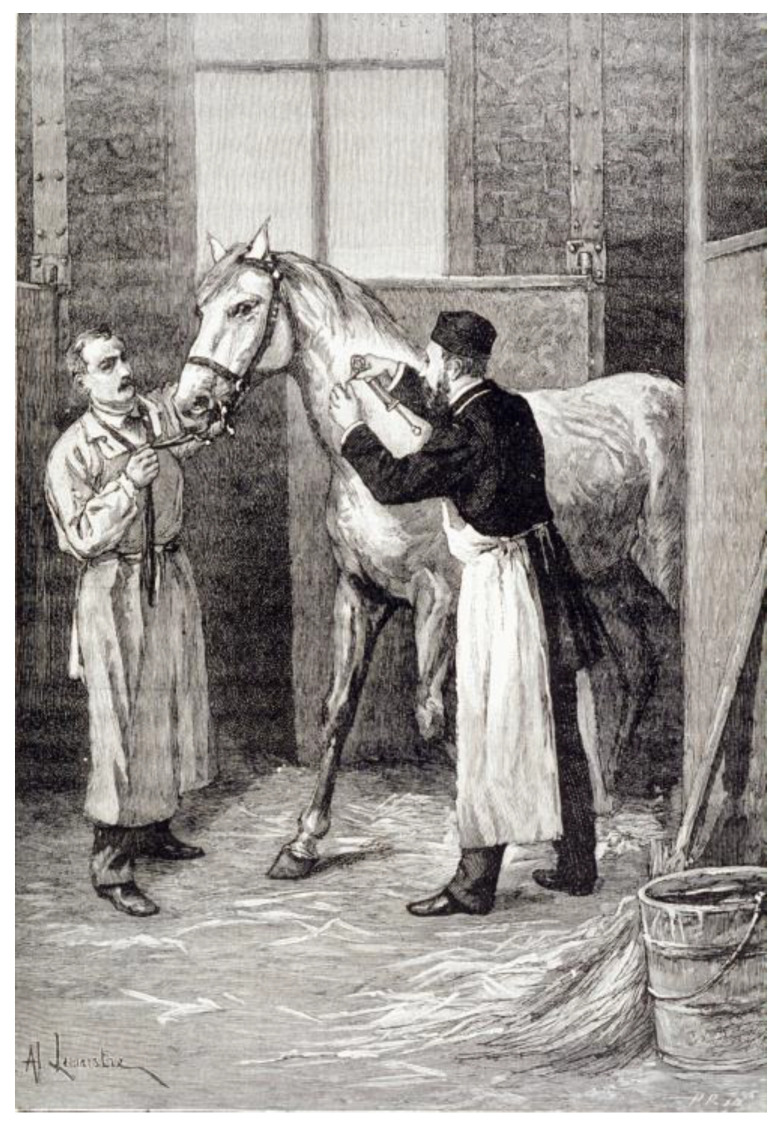
Inoculation of diphtheria toxin to horses producing antidiphtheria serum, at the stables of Marne la Coquette. Engraving after a drawing by Alexis Lemaistre (1896). ©Institut Pasteur/Musée Pasteur.

**Figure 3 toxins-14-00759-f003:**
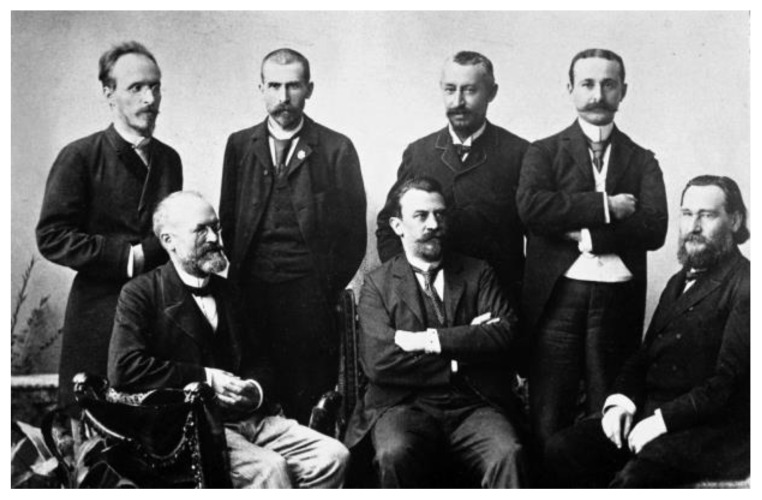
Members of the Budapest congress in 1894, lectures on the work on diphtheria. From left to right: Georges Gabritschevsky (1860–1907), Alphonse Laveran (1845–1922), Émile Roux (1853–1933), Léon Perdrix (1859–1917), Edmond Nocard (1850–1903), George Nuttal (1862–1937) and Elie Metchnikoff (1845–1916). ©Institut Pasteur/Musée Pasteur.

**Figure 4 toxins-14-00759-f004:**
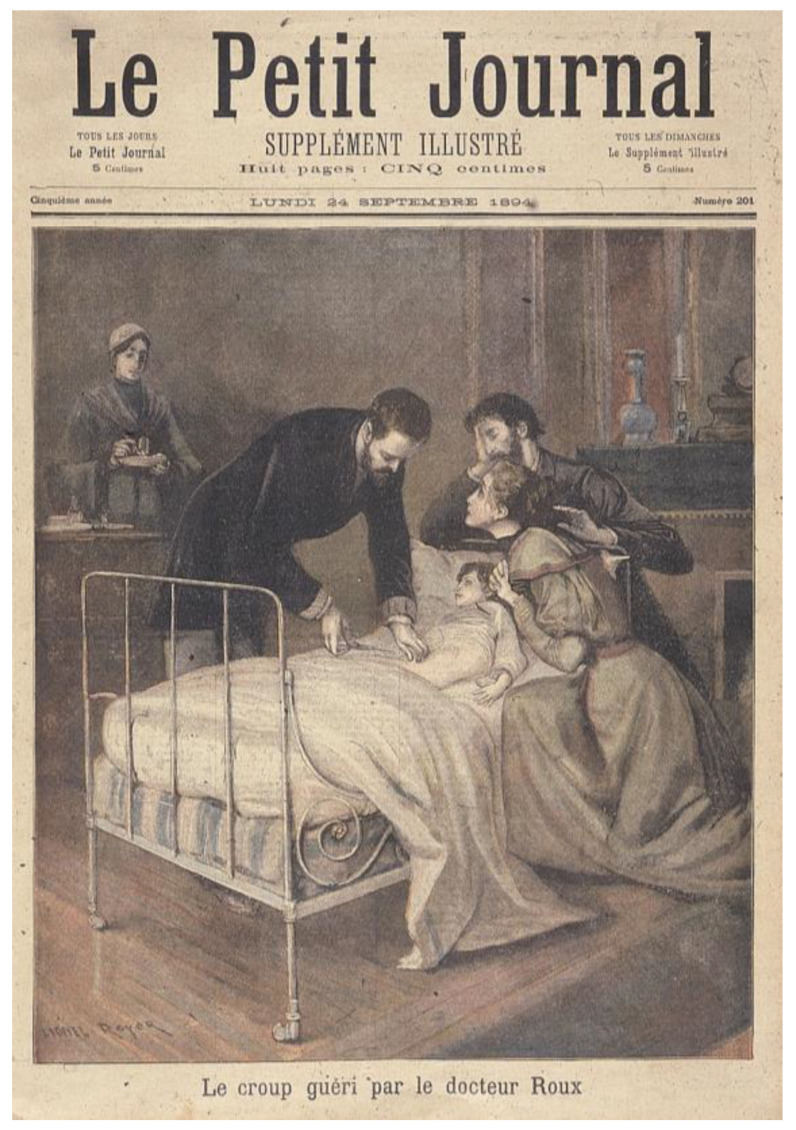
The front page of the “Petit Journal” (24 September 1894) showing Émile Roux saving a child from diphtheria thanks to serotherapy. ©Institut Pasteur/Musée Pasteur.

**Figure 5 toxins-14-00759-f005:**
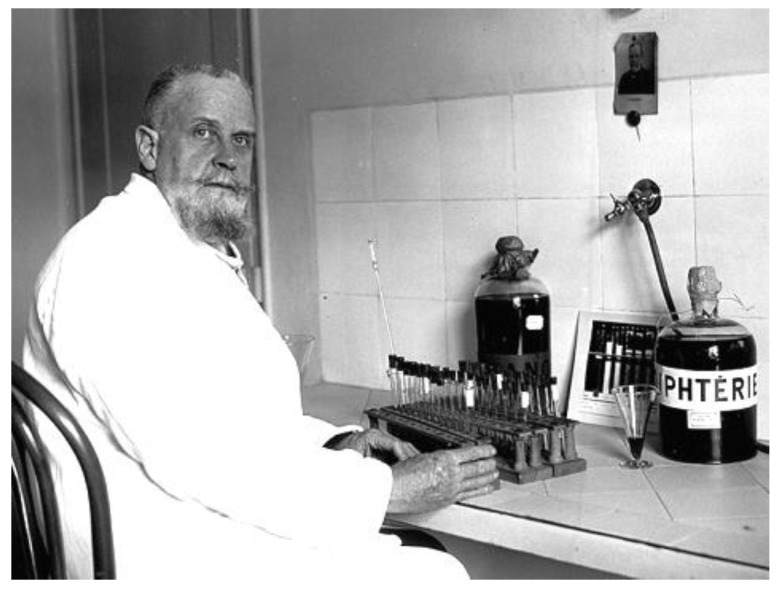
Gaston Ramon (1886–1963) who discovered the diphtheria anatoxin and the adjuvants ©Wikipedia.

**Figure 6 toxins-14-00759-f006:**
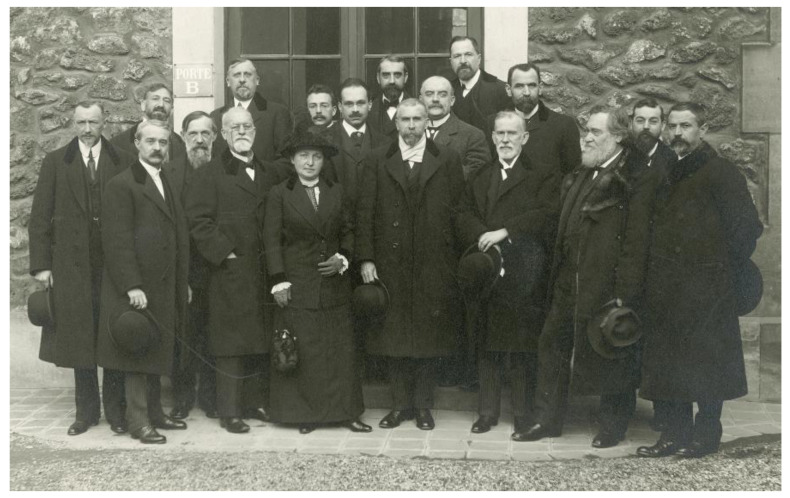
Visit of Paul Ehrlich and his wife at Institut Pasteur (4 February 1914). From left to right: Camille Delezenne, Constantin Levaditi, Salmon, Jean Danysz, Alphonse Laveran, Félix Mesnil, Mme Ehrlich, Eugène Wollman, Alexandre Besredka, Ernest Fourneau, Emile Roux, Auguste-Charles Marie, Alexandre Salimbeni, Paul Ehrlich, Louis Martin, Elie Metchnikoff, Gabriel Bertrand, Amédée Borrel. ©Institut Pasteur/Musée Pasteur.

## Data Availability

Not applicable.

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
