# Peer review of "From Bacterial Poisons to Toxins: The Early Works of Pasteurians"

_toxins, 2022, doi:10.3390/toxins14110759_

Round 1

Reviewer 1 Report

This review introduced the precursor works of the Pasteurian in the field of bacterial toxins. From this review, we can understand how Pasteurian contributes to the toxin and vaccine field.

It is also noted that the endotoxin study by Alexandre Besredka was introduced as the father of innate cellular immunity. It is an interest of the relationship between Metchnikoff and Besredka. Furthermore, it is also an interest in the relationship between Pateurians and German.

This paper is significant to know the history of toxin study not only in French but also Europe.

I recommend publishing it as an excellent review. 

Author Response

I thank the reviewer for his supportive and kind comments

Reviewer 2 Report

It is a fantastic work about the early history of bacterial toxins and is of great significance in memorizing the contributions of early scientists and learning their encourages to move the field of bacterial toxins forward.  while reading the article, it like went through the stories of important people in the history of microbiology and bacterial toxins, visualizing how the important discoveries were made by efforts and by luck.

there are only minor spelling errors for correction, here are some examples:

There are extra spaces in several lines of sentences. it is not clear either due to mistyping or covertion from word to PDF.

Line 14, between "and  Constant"

Extra spaces also are in the following sentences:

Lines 76, 90, 98, 158, 342,  353, 415, 

Line 37: "make a great contribution"

Line 44: "most fascinatingly"

Line 53, ", in which"

Line 71, "did not seem to"

Line 232, "155 folds or times"

Author Response

I thank the reviewer for his editing corrections. Those have been included in the revised manuscript

Reviewer 3 Report

The manuscript presents an interesting topic of science history namely the process of discovering that bacterial secreted products (we call them nowadays exotoxins or virulence factors or virulence-associated factors, respectively) are the pathogenic agents of bacterial infections and mostly not the bacteria themselves. Solidification of this concept took several decades during the 19th century, and the authors of this manuscript make an attempt to point out the achievements of French scientists associated with the Institut Pasteur in Paris (I guess that is what the authors mean by ‘Pasteurians’) in this context.

The manuscript presents an interesting perspective and indicates that French scientists have not always been properly honored for their scientific ideas and experimental achievements. However, this is, at the same time, a major drawback of this manuscript, in that it pushes aside the works of researchers of other nations who made very important contributions as well. This leaves the impression of a certain national bias in this science history piece which should have been avoided since such notions should get no room in science these days. The authors missed an opportunity to really put together a proper science history piece on the pathway from the discovery of the bacteria to the discovery that bacterial poisons or toxins are the major pathogenic principles in bacterial infections.

Another serious drawback of this manuscript is that hardly any original literature is cited but instead the authors refer to secondary or even tertiary publications. Many descriptions of results of important experiments or even literal quotes are not verified by citations at all (e.g., lines 41, 44, 46, 136-140, 302 etc.) leaving the impression that the compilation of this text has not been done very thoroughly.

A third point of major criticism is that the use of the English language is not appropriate in several parts of the text. For examples, I refer to lines 6 (bacteria cannot be the ‘ferments’ of putrefaction), 9 (‘ending to the concept’), 38 (time wrong), 64 (singular, plural), 69 (heading incomprehensible), etc. etc. 

Minor points:

  • The formatting of the text is faulty (different text styles, erratic use of italics).
  • Line 119 and elsewhere: The German researcher who was initially able to isolate and culture Corynebacterium diphteriawas Friedrich Loeffler (not Löffler!).
  • Line 149: Reference 21 is not correct at this position.
  • Line 213: The expression ‘very questionable Nobel Prize’ is not appropriate. The authors should at least explain the exact facts.
  • Lines 231 to 234: This is a nice anecdote, but irritating in the context of my first point of major criticism. Why do the authors mention this at all?
  • Lines 306 to 309: It is definitely not reasonable to deny Sambhu Nath De the recognition for the discovery that Vibrio cholerae secretes cholera toxin. If Metchnikoff has indeed drawn such a conclusion earlier than De, the authors should at least give a proper citation to that work.
  • Lines 338 and following: It is hard to comprehend what messages the authors want to transport with their Chapters 6 and 7. The information given is not really coherent and leave the impression of superficiality. I do not think that these chapters contribute anything essential to the main message of the manuscript. The description of Pasteur’s notion to call his optimized beer ‘Beer of the National Revenge’ is again a nice anecdote, but what does this really tell us other than that Pasteur was guided in some of his doings by national feelings. Quite understandable at that time but still incorrect (France started this war).

Author Response

Comments and Suggestions for Authors

The manuscript presents an interesting topic of science history namely the process of discovering that bacterial secreted products (we call them nowadays exotoxins or virulence factors or virulence-associated factors, respectively) are the pathogenic agents of bacterial infections and mostly not the bacteria themselves. Solidification of this concept took several decades during the 19th century, and the authors of this manuscript make an attempt to point out the achievements of French scientists associated with the Institut Pasteur in Paris (I guess that is what the authors mean by ‘Pasteurians’) in this context.

https://en.wiktionary.org/wiki/Pasteurian

The manuscript presents an interesting perspective and indicates that French scientists have not always been properly honored for their scientific ideas and experimental achievements. However, this is, at the same time, a major drawback of this manuscript, in that it pushes aside the works of researchers of other nations who made very important contributions as well. This leaves the impression of a certain national bias in this science history piece which should have been avoided since such notions should get no room in science these days. The authors missed an opportunity to really put together a proper science history piece on the pathway from the discovery of the bacteria to the discovery that bacterial poisons or toxins are the major pathogenic principles in bacterial infections.

I am sorry that the reviewer had this unpleasant feeling. Of note, another reviewer wrote: “This paper is significant to know the history of toxin study not only in France but also Europe.”. This paper was an invited paper for a special issue aimed to celebrate the bicentenary of the birth of Louis Pasteur, and I was asked to focus on the work of the Pasteurians. A more comprehensive review on the main European contributors who were involved in the discoveries of toxins can be find at : Cavaillon, J-M. Historical links between toxinology and immunology Pathogens and Disease 2018, 76, fty019. Nevertheless, in the present  review the works of Panum, Bergmann, Selmi, Brieger, Gamaleïa, Ruffer, Pfeiffer, Klebs, Loeffler, Faber were mentioned. To satisfy the reviewer, I am now also mentioning the contribution of Centanni, and Cantani.

Another serious drawback of this manuscript is that hardly any original literature is cited but instead the authors refer to secondary or even tertiary publications. Many descriptions of results of important experiments or even literal quotes are not verified by citations at all (e.g., lines 41, 44, 46, 136-140, 302 etc.) leaving the impression that the compilation of this text has not been done very thoroughly.

Among the previous 55 references, 44 corresponded to original citations. I have added few new ones:

  • Bergmann, E.V. ; Schmiedeberg O. Ueber dasschwefelsaure sepsin (das gift faulender substanzen). Wisschenschaften 1866, 32, 497–8 
  • Brieger L. Zur kenntniss des tetanin und des mytilotoxin. Pathol. Anat. 1888, 112, 549–51 
  • Selmi, F. Sulle ptomaine od alcaloidi cadaverici e loro importanza in tossicologia, osservazioni del prof. Francesco Selmi aggiuntavi una perizia per la ricerca della morfina. Bologna: N. Zanichelli, 1878 
  • Fibiger, J. Om serumbehandling af difteri. Hospitalstidende 1898; 6: 309–325, 337–50
  • Bordet J. Les leucocytes et les propriétés actives du sérum chez les vaccinés. Inst. Pasteur. 1895, 9, 462–506 
  • Boivin, A., Mesrobeanu, I., Mesrobeanu, L. Technique pour la préparation des polysaccharides microbiens spécifiques. R. Soc. Biol. 1933, 113, 490–492

Line 41: reference 8 was called.

Line 44: the reference: Pfeiffer R. Untersuchungen über das Choleragift. Zeitschr Hygiene 1892;11:393–412 has been added.

Line 46: A review is now referenced: Dinarello  C.A. Cytokines as Endogenous Pyrogens. Infect. Diseases 1999, 179, S294-S304.

Line 136-140: The quotation corresponds to the previous reference #20 (now ref. #26).

Line 302: The description of Metchnikoff’s work corresponds to the previous reference #46 (now #55) called initially line 293

A third point of major criticism is that the use of the English language is not appropriate in several parts of the text. For examples, I refer to lines 6 (bacteria cannot be the ‘ferments’ of putrefaction), 9 (‘ending to the concept’), 38 (time wrong), 64 (singular, plural), 69 (heading incomprehensible), etc. etc. 

lines 6 (bacteria cannot be the ‘ferments’ of putrefaction): despite is was the concept of Pasteur, it has been replaced by  “bacteria as the cause of putrefaction

Line 9 (‘ending to the concept’): replaced by  “resulting in the concept

Line 38 (time wrong): “will become” changed to “became

64 (singular, plural):  “infectious diseases were an intoxication by the poisons of the pathogenic microbes” changed to  “infectious diseases were intoxications by the poisons of the pathogenic microbes”

Line 69 (heading incomprehensible): “Pasteur, the soluble chemical products produced by bacteria and immunity” changed to “The soluble chemical products produced by bacteria and immunity”

Minor points:

  • The formatting of the text is faulty (different text styles, erratic use of italics).

This has been corrected, although some italic parts were wrong modifications made during the conversion to the pdf format

  • Line 119 and elsewhere: The German researcher who was initially able to isolate and culture Corynebacterium diphteria was Friedrich Loeffler (not Löffler!).

We have now adopted the spelling suggested by the reviewer while both can be used

  • Line 149: Reference 21 is not correct at this position.

Sorry, reference 22 should have been called there.

  • Line 213: The expression ‘very questionable Nobel Prize’ is not appropriate. The authors should at least explain the exact facts.

A reference is now given to support the statement

Disproof of FIBIGER's discovery [FROM WIKIPEDIA]

Experimental evidence later refuted Fibiger's Nobel Prize discovery. In 1918, there was a critical comment from F. D. Bullock and G. L. Rohdenburg that Fibiger could have confused cancer-like (neoplastic) tumour from true (metastatic) cancer, and that he had not induced actual cancer.[17] But Fibiger responded "That these tumors are true carcinomata cannot, thus, be doubted, and the fact that they may occur in younger animals does not diminish our right to range them among the true malignant neoplasms."[3] After his death in 1928, there was a better understanding of the nature of cancer—on the differences of neoplastic tumour and malignant tumours (cancer), challenging the claims of Fibiger. The most important criticism was by Richard Douglas Passey, with his colleagues A. Léese, and J.C. Knox. Their experimental conclusions were that the nematode could not cause cancer, and that experimentally-induced cancer was due to other factors such as vitamin A deficiency.[18] Fibiger used a vitamin A-less diet for his experimental rats, and it was by then established that vitamin A deprivation alone could induce cancer.[19][20][21] An elaborate experiment by W. Cramer in 1937 came to the conclusion that Fibiger's tumour could not be a true cancer.[22] A more rigorous study was done by Claude R. Hitchcock and E. T. Bell in 1952. They repeated Fibiger's experiments using advanced techniques, and concluded that the tumours induced by the nematode in rats were metaplasia, not cancer. All tumours were due to vitamin A deficiency.[23] Systematic reanalysis of Fibiger's data also gave the same conclusion that G. neoplasticum can cause tumour, but is not carcinogenic by itself.[24][25]

  • Lines 231 to 234: This is a nice anecdote, but irritating in the context of my first point of major criticism. Why do the authors mention this at all?

I consider that that these high numbers of nominations for the Nobel Prize further illustrate the importance of the work of Ramon and Roux recognized by their contemporary peers.

  • Lines 306 to 309: It is definitely not reasonable to deny Sambhu Nath De the recognition for the discovery that Vibrio cholerae secretes cholera toxin. If Metchnikoff has indeed drawn such a conclusion earlier than De, the authors should at least give a proper citation to that work.

I do not deny the work of Sambhu Nath De, I just mentioned that Metchnikoff was among the first to address the presence of a toxin produced by Vibrio cholera. To be fully correct I now pay tribute to the precursor works on cholera toxin of Antonio Cantani in 1886, and Nikolaï Gamaleïa in 1892.

The previous quoted citation of S.N De has been replaced by the proper one : De, S. N., Sarkar, J. K., Tribedi, B. P. An experimental study of the action of cholera toxin. J. Pathol. Bacteriol. 63: 707–717, 1951

  •  Lines 338 and following: It is hard to comprehend what messages the authors want to transport with their Chapters 6 and 7. The information given is not really coherent and leave the impression of superficiality. I do not think that these chapters contribute anything essential to the main message of the manuscript. The description of Pasteur’s notion to call his optimized beer ‘Beer of the National Revenge’ is again a nice anecdote, but what does this really tell us other than that Pasteur was guided in some of his doings by national feelings. Quite understandable at that time but still incorrect (France started this war).

The aim of these chapters was (i) to illustrate that Pasteurians contributed to the early publications of books devoted to toxins, (ii) and to illustrate that despite the Germanophobia of Louis Pasteur, his close collaborators did not follow him and most importantly contributed to a respectful interaction with their German colleagues.

I would like to maintain these chapters regarding the laudatory comments of the three other reviewers.

Reviewer 4 Report

The historical review very well written and illustrative. I have no concerns or additional comments.  

Author Response

I thank the reviewer for his support.

Round 2

Reviewer 3 Report

In my note to the editor accompanying my review of the first version of this manuscript I wrote: "I think that this manuscript might be suitable as a commemorative for some anniversary of the Institut Pasteur, ..."That exactly this was the intention of the article as the authors state in their responses was not known to me at that time. In consequence, I now come to a different conclusion than in my first review.

However, there are still some points that need mending:

Citations: Due to some additions in the reference list, the entire numbering system got messed up. This should be carefully revised.

l. 126-128: Sentence is incomprehensible.

l. 130: deaths

l. 175: cities

l. 185-187: Sentence is very twisted.

l. 215-219: I still insist that the term 'very questionable Nobel Prize' is not appropriate because this term would denounce the competence of the members of the Nobel committee. As it occurs in science (and this is science in itself) it turned out later that the scientific results for which the Nobel prize had been given to Fibiger were not entirely correct. The authors pasted the Wikipedia article about this in their comments, but the explanation they give in the text 1) does not support their statement, and 2) is still inappropriate. Moreover, it is wrong, as a substance that causes tumors is definitely a carcinogen. This whole point should therefore be reworded.

l. 234: titer

l. 264: eleven years old

l. 270: insensitive

l. 322: endotoxins

l. 330: remove comma

l. 261: animal venoms?

l. 384: came in charge

l. 393: American Thermo Electron

l. 417: loss

Reference list: Formating of many citations is faulty.

Ref. 26: There three bibliographic sources given for one article. How that?

Ref. 30, spelling: Immunität

Author Response

In my note to the editor accompanying my review of the first version of this manuscript I wrote: "I think that this manuscript might be suitable as a commemorative for some anniversary of the Institut Pasteur, ..."That exactly this was the intention of the article as the authors state in their responses was not known to me at that time. In consequence, I now come to a different conclusion than in my first review.

However, there are still some points that need mending:

Citations: Due to some additions in the reference list, the entire numbering system got messed up. This should be carefully revised.

I thank the reviewer for his/her careful reading. Indeed, ref#41 has been changed to #42, ref. #53 changed to 52 and ref#45 to #53

  1. 126-128: Sentence is incomprehensible.

The sentence has been rephrased

  1. 130: deaths
  2. 175: cities

Thank you for your editing

  1. 185-187: Sentence is very twisted.

The sentence has been rephrased

  1. 215-219: I still insist that the term 'very questionable Nobel Prize' is not appropriate because this term would denounce the competence of the members of the Nobel committee. As it occurs in science (and this is science in itself) it turned out later that the scientific results for which the Nobel prize had been given to Fibiger were not entirely correct. The authors pasted the Wikipedia article about this in their comments, but the explanation they give in the text 1) does not support their statement, and 2) is still inappropriate. Moreover, it is wrong, as a substance that causes tumors is definitely a carcinogen. This whole point should therefore be reworded.

To satisfy the reviewer, I have I softened my words, despite the numerous articles published from 1935 (Passey et al.  J. Pathol. Bacteriol. 1935, 40 , 198–199) to nowadays (Lichtman MA. Rambam Maimonides Med J. 2022 Jul 31;13(3):e0022) claiming that Fibiger’s finding was found to be erroneous.

It is out of question to denounce the competence of the Nobel Committee. However, this committee is composed of human beings and those can make mistakes. Although it was very rare, it happened: e.g. Egas Moniz for the lobotomy; Julius Wagner-Jauregg for the malaria therapy. Furthermore, even sometimes the integrity of Nobel laureates can be questioned (Holly Else. Dozens of papers co-authored by Nobel laureate raise concerns  doi: 10.1038/d41586-022-03032-9)!

  1. 234: titer
  2. 264: eleven years old
  3. 270: insensitive
  4. 322: endotoxins
  5. 330: remove comma
  6. 261: animal venoms?
  7. 384: came in charge
  8. 393: American Thermo Electron
  9. 417: loss

Thank you for your editing. All corrections have been made.

Reference list: Formating of many citations is faulty.

corrections made.

Ref. 26: There three bibliographic sources given for one article. How that?

There is only one ref. #26 that corresponds to the different quotations

Ref. 30, spelling: Immunität

Thank you for your editing